# Comparable Effectiveness of Cefuroxime and Piperacillin-Tazobactam as Empirical Therapy for Methicillin-Susceptible *Staphylococcus aureus* Bacteremia

Robert Strengen Bigseth,[a,b] Håkon Sandholdt,[a] Andreas Petersen,[c] Christian Østergaard,[d] Thomas Benfield,[a,b] Louise Thorlacius-Ussing[a,b]

[a]CREDID (Center of Research & Disruption of Infectious Diseases), Department of Infectious Diseases, Copenhagen University Hospital, Amager and Hvidovre, Hvidovre, Denmark
[b]Institute of Clinical Medicine, Faculty of Health and Medical Sciences, University of Copenhagen, Copenhagen, Denmark
[c]Reference Laboratory for Antimicrobial Resistance, Statens Serum Institut, Copenhagen, Denmark
[d]Department of Clinical Microbiology, Copenhagen University Hospital, Amager and Hvidovre, Hvidovre, Denmark

**ABSTRACT**   Our objective was to examine whether empirical antimicrobial therapy (EAT) against methicillin-susceptible *Staphylococcus aureus* bacteremia (MS-SAB) with piperacillin-tazobactam (TZP), cefuroxime or combination therapy with one of these was differentially associated with 7-, 30-, and 90- day all-cause mortality or MS-SAB relapse. A multicenter retrospective cohort study of adults with MS-SAB from 2009 through 2018 was used, and 7-, 30-, 90-day mortality and relapse within 90 days were assessed and expressed as hazard ratio (HR) with a 95% confidence interval (95% CI) using Cox proportional hazard regression analysis. Matching of the two monotherapy groups was performed using propensity score matching. In total, 1158 MS-SAB cases were included and received one of three EAT regimens: TZP (*n* = 429), cefuroxime (*n* = 337), or TZP or cefuroxime with one or more additional effective antimicrobial (*n* = 392). The overall 30-day mortality was 28.0% (25.5 to 30.3%). After adjustment and matching, there was no significant difference in 7-, 30-, or 90-day mortality between the therapy groups. The matched HR of death was 0.81 (95% CI, 0.38 to 1.76) at 7 days, 0.82 (95% CI, 0.47 to 1.46) at 30 days, and 0.81 (95% CI, 0.50 to 1.32) at 90 days for TZP compared with cefuroxime. Adjusted HR of 90-day relapse was insignificant between the three therapy groups: TZP: 1.55 (95% CI, 0.54 to 4.43); combination therapy: 1.73 (95% CI, 0.62 to 4.80) compared to cefuroxime. There was no significant difference in 7-, 30-, or 90-day mortality or relapse between MS-SAB patients treated with empirical TZP or cefuroxime after adjustment and matching of covariables.

**IMPORTANCE** This multicenter retrospective matched cohort study evaluated the effect of empirical antimicrobial therapy on the clinical outcome of methicillin-susceptible *Staphylococcus aureus* bacteremia (MS-SAB) in >1100 adult patients. To the best of our knowledge, this is the largest study to date evaluating the effect of empirical treatment on the MS-SAB outcome. Importantly, the study found no significant difference in either short- or long-term mortality nor relapse between patients with MS-SAB receiving empirical treatment with cefuroxime or piperacillin-tazobactam. As such, this study provides crucial contemporary data supporting the widespread clinical practice of initiating empirical antimicrobial therapy of sepsis with $\beta$-lactam-$\beta$-lactamase-inhibitor.

**KEYWORDS** *S. aureus*, bacteremia, sepsis, empirical therapy, piperacillin-tazobactam, pip-tazo, cefuroxime, combination therapy, MS-SAB, MSSA

Address correspondence to Robert Strengen Bigseth, robert.bigseth@outlook.com.

The authors declare no conflict of interest.

Methicillin-susceptible *Staphylococcus aureus* bacteremia (MS-SAB) is one of the foremost causes of Gram-positive bacteremia and therefore an important cause of sepsis (1). Although the incidence of methicillin-resistant SAB (MR-SAB) increased

during the last decade, more contemporary data have found decreasing rates of MR-SAB and MS-SAB are still the predominant cause of SAB in many western countries (2–4). Empirical antimicrobial therapy (EAT) is the cornerstone of sepsis therapy during the initial phase. Delay in time to appropriate antimicrobial therapy is correlated with sepsis mortality in general (5). A similar association has been demonstrated for nosocomial SAB (6). Combination therapy is often used for more sick patients and difficult-to-treat infections because it is perceived that clearance of *S. aureus* likely is improved, although clinical evidence to support this notion is limited (7).

Cefuroxime, a second-generation cephalosporine, is recommended as empirical therapy drug against a wide range of severe infections, especially in cases of penicillin allergy (8). Few studies have investigated the effect of cefuroxime compared to other cephalosporins against MS-SAB and have reported inconsistent findings (9–11). In Denmark, cefuroxime is the most frequently used cephalosporin against MS-SAB. Piperacillin-tazobactam (TZP), a $\beta$-lactam-$\beta$-lactamase-inhibitor (BLBLI), is another widely used EAT in sepsis.

In the past decade, antimicrobial stewardship in our hospital network and elsewhere has advised against the use of cephalosporins and quinolones due to the emergence of extended-spectrum beta-lactamase (ESBL) producing Enterobacteriaceae and an epidemic of *Clostridioides difficile* infection (12–16). A transition from the use of cefuroxime to TZP as the preferred empirical antimicrobial gradually occurred from 2010 to 2015 (12, 16). There are, however, some studies that have reported an increased risk of mortality in patients with MS-SAB treated with TZP compared to cephalosporins (9, 17). The sample size of these studies has been limited ($n < 600$), and as such, further investigations are needed to conclude this matter. In addition, there is a concern that *S. aureus* may exhibit an inoculum effect on TZP, i.e., the MIC of TZP is elevated as the number of organisms increases (18).

The choice of EAT for sepsis is not based on evidence from controlled trials but rather extrapolations of *in vitro* data, animal models, epidemiology, and clinical experience. In the absence of randomized controlled trial data, we presented data from a large cohort study after careful adjustment and matching of covariables to minimize the risk of confounding by indication. The objective was to examine if there was a differential effect of TZP and cefuroxime on 7-, 30-, and 90-day all-cause mortality or relapse associated with MS-SAB.

## RESULTS

Of the 1969 cases of MS-SAB, 168 (8.5%) did not receive any EAT, 21 (1.1%) received ineffective EAT, and 622 (31.6%) received other regimens than those studied here (Fig. 1). Of the included individuals, 429 (37.0%) received piperacillin-tazobactam, 337 (29.1%) received monotherapy with cefuroxime, and 392 (33.9%) were treated with a combination of cefuroxime or TZP and at least one other anti-staphylococcal drug. Details on the combination therapy group are in Table S2.

Characteristics of the three therapy groups are listed in Table 1. The two monotherapy groups were similar in terms of age and sex while the combination therapy group was younger and more often men.

Cases in the TZP monotherapy group were more likely to have a higher Charlson Comorbidity Index (CCI) score compared to the other groups (OR 1.88 [95% CI, 1.30 to 2.73] for TZP compared to cefuroxime). The indication to initiate EAT differed between the three treatment groups: "skin, soft tissue, or bone infection" was more often associated with administration of cefuroxime alone (OR 2.79 [95% CI, 1.80 to 4.32] for cefuroxime compared to TZP) while TZP alone was associated with a lack of a specifically stated indication. Significant differences regarding the primary focus of the infections were observed between the therapy groups. As such, skin and postoperative infections were more frequent in the cefuroxime therapy group. Contrary, the TZP and the combination therapy group had a higher proportion of cases with a primary respiratory/pulmonary focus of the infection compared to the cefuroxime therapy group. The rate of cases with an unknown focus of infection was similar for the two monotherapy

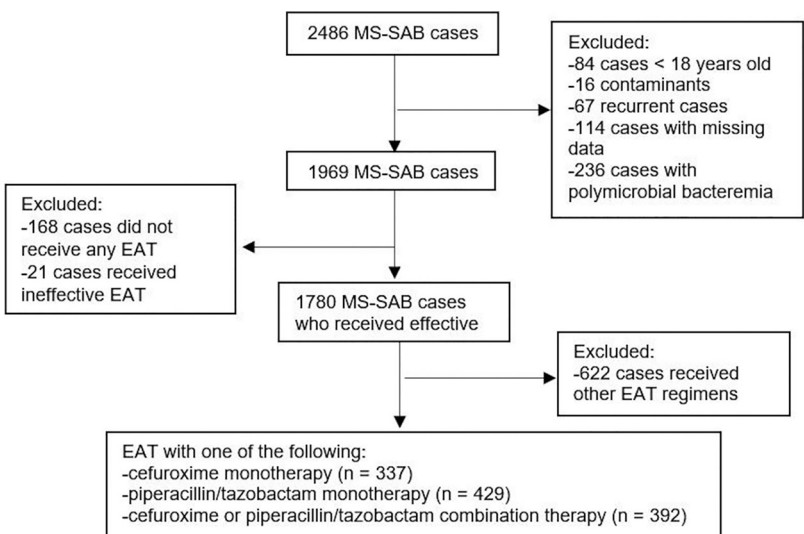

**FIG 1** Flow chart of methicillin-susceptible *S. aureus* bacteremia cases.

groups. Patients in the TZP and the combination therapy group were more likely to undergo echocardiography, both transthoracic (TTE) and transesophageal (TEE), compared to the cefuroxime group.

The median duration of EAT was significantly longer in the cefuroxime monotherapy group (2.5 days; IQR, 1 to 6) compared to the TZP monotherapy group (1 day; IQR, 1 to 2) and the combination therapy group (1.5 days; IQR, 1 to 3). Also, the choice of definitive antimicrobial therapy differed between the therapy groups. A higher proportion of patients in the cefuroxime group (50.1%) received unspecified definitive antimicrobial therapy, including combination therapy compared to the other groups (TZP: 22.1%; combination therapy: 43.1%). In the TZP group, 9.8% of the patients received cefuroxime as definitive therapy.

**Mortality.** No significant difference in 7- or 30-day mortality between the three therapy groups was found in the crude or adjusted analyses (Table 2). In the crude model, 90-day mortality was significantly higher for patients who received TZP monotherapy as EAT (HR 1.33 [95% CI, 1.06 to 1.68]) compared to patients who received cefuroxime monotherapy. However, after adjustment for covariables this was no longer statistically significant (HR 0.98 [95% CI, 0.71 to 1.37]). Similarly, unadjusted HRs after 7, 30, and 90 days were higher for the TZP monotherapy group than the combination therapy group, but not after adjustment (Table 2 and Fig. 2A).

Analyses on separate combination therapy groups, including either TZP or cefuroxime were performed and did not show any significant differences in 7-, 30-, or 90-day mortality or relapse in crude or adjusted analyses compared to cefuroxime monotherapy (Table S3).

Of the 766 cases treated with either TZP or cefuroxime alone, 451 (58.9%) cases were matched (Table S4). Similar to the unmatched analysis, treatment with TZP monotherapy had a comparable HR of mortality to treatment with cefuroxime monotherapy (7-d HR 0.81 [95% CI, 0.38 to 1.76], 30-d HR 0.82 [95% CI, 0.47 to 1.46] and 90-d mortality HR 0.81 [95% CI, 0.50 to 1.32]; Table 3).

Separate analyses stratified by treatment duration on patients who received cefuroxime or TZP as empirical monotherapy ≥3 days or <3 days in total are shown in Table 4 and 5. In the subgroup of short EAT duration, results were insignificant between the two therapy groups except for the adjusted 30-day mortality for TZP. Contrary, in patients receiving ≥3 days of EAT, TZP monotherapy was consistently associated with higher crude and adjusted HR of death after 7, 30, and 90 days. However, in the matched

**TABLE 1** Characteristics of methicillin-susceptible *Staphylococcus aureus* bacteremia cases stratified by empirical antimicrobial therapy

| Characteristic | Cefuroxime monotherapy (*n* = 337) | Piperacillin-tazobactam monotherapy (*n* = 429) | Combination therapy with cefuroxime or piperacillin-tazobactam (*n* = 392) | P value |
|---|---|---|---|---|
| Female (%) | 140 (41.5) | 175 (40.8) | 133 (33.9) | 0.12 |
| Median age (IQR[a]) | 73.0 (61.0-84.0) | 75.0 (61.0-84.0) | 70.0 (58.0-80.2) | 0.0044 |
| Comorbidity score (CCI) | | | | |
| Low, CCI = 0 (%) | 79 (23.4) | 60 (14.0) | 68 (17.3) | |
| Medium, CCI = 1-2 (%) | 123 (36.5) | 173 (40.3) | 171 (43.6) | |
| High, CCI > 2 (%) | 135 (40.1) | 196 (45.7) | 153 (39.0) | < 0.001 |
| Smoking (%) | 79 (23.4) | 107 (24.9) | 98 (25.0) | 0.93 |
| Daily alcohol consumption (%) | 59 (17.5) | 108 (25.2) | 105 (26.8) | 0.019 |
| Injection drug use (%) | 9 (2.7) | 15 (3.5) | 10 (2.6) | 0.077 |
| Any immunosuppression[b] (%) | 20 (5.9) | 21 (4.9) | 34 (8.7) | 0.24 |
| Echocardiography | 193 (57.3) | 294 (68.5) | 262 (66.8) | 0.0075 |
| Transthoracic echocardiography, TTE (%) | 163 (48.4) | 263 (61.3) | 226 (57.7) | 0.0025 |
| Transesophageal echocardiography, TEE (%) | 67 (19.9) | 103 (24.0) | 96 (24.5) | 0.33 |
| Infectious disease specialist consultation | 63 (18.7) | 73 (17.0) | 77 (19.6) | 0.17 |
| Penicillin-susceptible *S. aureus* (PSSA) isolate | 64 (19.0) | 99 (23.1) | 81 (20.7) | 0.16 |
| Acquisition of bacteremia | | | | |
| Community-acquired (%) | 181 (53.7) | 223 (52.0) | 221 (56.4) | |
| Nosocomial (%) | 101 (30.0) | 141 (32.9) | 111 (28.3) | |
| Healthcare-acquired (%) | 49 (14.5) | 62 (14.5) | 55 (14.0) | |
| Unknown (%) | 6 (1.8) | 3 (0.7) | 5 (1.3) | 0.66 |
| Primary focus | | | | |
| IV device infection (%) | 30 (8.9) | 43 (10.0) | 37 (9.4) | |
| Dialysis (%) | 6 (1.8) | 5 (1.2) | 14 (3.6) | |
| Skin infection (%) | 44 (13.1) | 46 (10.7) | 35 (8.9) | |
| Respiratory infection (%) | 17 (5.0) | 36 (8.4) | 37 (9.4) | |
| Urinary tract infection (%) | 7 (2.1) | 11 (2.6) | 20 (5.1) | |
| Postoperative infectiogn (%) | 15 (4.5) | 9 (2.1) | 13 (3.3) | |
| Prothesis infection (%) | 22 (6.5) | 19 (4.4) | 22 (5.6) | |
| Other focus (%) | 32 (9.5) | 43 (10.0) | 41 (10.5) | |
| Unknown focus (%) | 164 (48.7) | 217 (50.6) | 173 (44.1) | < 0.001 |
| Median SOFA score at onset (IQR) | 2 (1-4) | 3 (1-4) | 3 (1-5) | 0.056 |
| Pitt score ≥ 4 at onset (%) | 18 (6.8) | 22 (6.1) | 34 (10.5) | 0.048 |
| Indication of EAT | | | | |
| Fever with unknown focus (%) | 82 (24.3) | 123 (28.7) | 83 (21.2) | |
| Skin, soft tissue or bone infection (%) | 67 (19.9) | 35 (8.2) | 58 (14.8) | |
| Urinary tract infection (%) | 35 (10.4) | 55 (12.8) | 56 (14.3) | |
| IV device infection (%) | 10 (3.0) | 11 (2.6) | 18 (4.6) | |
| Pneumonia (%) | 82 (24.3) | 104 (24.2) | 82 (20.9) | |
| Not mentioned (%) | 36 (10.7) | 74 (17.2) | 53 (13.5) | |
| Other (%) | 25 (7.4) | 27 (6.3) | 42 (10.7) | < 0.001 |
| Duration of EAT, median days (IQR) | 2.5 (1-6) | 1 (1-2) | 1.5 (1-3) | < 0.001 |
| Duration of definitive therapy, median days (IQR) | 14 (7.8-27.2) | 14 (8.0-24.0) | 15 (9.0-28.8) | 0.020 |
| Definitive therapy drug[c] | | | | |
| Dicloxacillin monotherapy (%) | 146 (43.3) | 255 (59.4) | 173 (44.1) | |
| Penicillin monotherapy (%) | 16 (4.7) | 37 (8.6) | 19 (4.8) | |
| Cefuroxime monotherapy (%) | 6 (1.8) | 42 (9.8) | 31 (7.9) | |
| Other, incl. comb. therapy (%) | 169 (50.1) | 95 (22.1) | 169 (43.1) | |
| Any secondary manifestation (%) | 105 (31.2) | 131 (30.5) | 163 (41.6) | 0.0039 |
| Endocarditis (%) | 16 (4.7) | 41 (9.6) | 50 (12.8) | < 0.001 |
| Osteomyelitis (%) | 15 (4.5) | 14 (3.3) | 17 (4.3) | 0.54 |
| Spondylodiscitis (%) | 19 (5.6) | 25 (5.8) | 19 (4.8) | 0.65 |
| Arthritis (%) | 16 (4.7) | 16 (3.7) | 21 (5.4) | 0.23 |
| Meningitis (%) | 0 (0.0) | 2 (0.5) | 6 (1.5) | 0.023 |
| Pneumonia (%) | 25 (7.4) | 30 (7.0) | 40 (10.2) | 0.21 |

**TABLE 1** (Continued)

| Characteristic | Cefuroxime monotherapy (*n* = 337) | Piperacillin-tazobactam monotherapy (*n* = 429) | Combination therapy with cefuroxime or piperacillin-tazobactam (*n* = 392) | *P* value |
|---|---|---|---|---|
| Other (%) | 28 (8.3) | 17 (4.0) | 39 (9.9) | 0.0090 |
| Relapse within 90 days (%) | 8 (2.4) | 21 (4.9) | 18 (4.6) | 0.27 |
| 7-day mortality (%) | 47 (13.9) | 71 (16.6) | 55 (14.0) | 0.030 |
| 30-day mortality (%) | 86 (25.5) | 130 (30.3) | 110 (28.1) | 0.034 |
| 90-day mortality (%) | 115 (34.1) | 186 (43.4) | 141 (36.0) | < 0.001 |

[a]IQR, interquartile range.
[b]HIV positive, chemotherapy, other immunosuppressive treatment, or other immunosuppression.
[c]Defined as therapy on day three after blood culture yielding.

analyses, TZP was not significantly associated with the clinical outcome (Table 5). Characteristics of the subgroups are provided in Table S5.

**Relapse.** The incidence of MS-SAB relapses between the therapy groups was comparable (*P* = 0.27), ranging from 2.4% in the cefuroxime monotherapy group to 4.9% in the TZP monotherapy group (Fig. 2B). Crude and adjusted HR of relapse were similar for all three therapy groups (TZP: crude HR 2.08 [95% CI, 0.92 to 4.71], adjusted HR 1.55 [95% CI, 0.54 to 4.43]; combination therapy: crude HR 1.95 [95% CI, 0.85 to 4.48], adjusted HR 1.73 [95% CI, 0.62 to 4.80] compared to cefuroxime).

As seen in Fig. 3, a higher relative proportion of the patients received cefuroxime as EAT in the earlier years of the study period compared to recent years. The figure also shows a gradual increase in the proportion of patients who receive TZP monotherapy as EAT from 2011 until 2017.

## DISCUSSION

This multicenter study did not find any significant differences in the clinical outcomes from MS-SAB between patients treated with empirical piperacillin-tazobactam or cefuroxime with or without combination antimicrobials. This large cohort provides contemporary data on 7-, 30-, and 90-day mortality or relapse to support the widespread clinical practice of initiating empirical antimicrobial therapy for sepsis with a $\beta$-lactam-$\beta$-lactamase-inhibitor.

Previous studies on this matter have been contradictory (9, 17, 19, 20). Forsblom et al. (20) found no significant difference in 28-day and 90-day mortality in first-week monotherapy of MS-SAB with cloxacillin compared with continuation of empirical cefuroxime or ceftriaxone. In contrast, Paul et al. (9) reported higher 30-day mortality of MS-SAB after 2 days of empirical monotherapy with cefuroxime, ceftriaxone, cefotaxime, or BLBLIs compared with cloxacillin or cefazolin. However, the study was not matched, which increases the risk of confounding by indication. In addition, a therapy group of BLBLIs, in general, was defined, which included amoxicillin-clavulanic acid, ampicillin/sulbactam, and TZP, making it difficult to conclude any specific therapeutic effect of TZP *per se*. Recently, Beganovic et al. (17) reported higher 30-day mortality of patients with MS-SAB exclusively treated with TZP compared to nafcillin, oxacillin, or cefazolin. This study did not examine empirical therapy specifically. Besides, it only

**TABLE 2** Crude and adjusted hazard ratios of patients with methicillin-susceptible *S. aureus* bacteremia receiving empirical therapy with cefuroxime, piperacillin-tazobactam, or combination therapy

| Empirical therapy | 7-day mortality | | 30-day mortality | | 90-day mortality | |
|---|---|---|---|---|---|---|
| | Crude HR[a] (95 % CI) | Adjusted HR (95 % CI) | Crude HR (95 % CI) | Adjusted HR (95 % CI) | Crude HR (95 % CI) | Adjusted HR (95 % CI) |
| Cefuroxime monotherapy (*n* = 337) | 1.00 | 1.00 | 1.00 | 1.00 | 1.00 | 1.00 |
| Piperacillin-tazobactam monotherapy (*n* = 429) | 1.19 (0.83-1.73) | 1.03 (0.60-1.75) | 1.22 (0.93-1.60) | 1.09 (0.71-1.66) | 1.33 (1.06-1.68) | 0.98 (0.71-1.37) |
| Combination therapy (*n* = 392) | 0.99 (0.67-1.47) | 1.02 (0.62-1.67) | 1.10 (0.83-1.46) | 1.10 (0.74-1.64) | 1.06 (0.83-1.36) | 1.12 (0.82-1.53) |

[a]HR, hazard ratio; CI, confidence interval.

**TABLE 3** Adjusted and propensity score-matched hazard ratios of patients with methicillin-susceptible *S. aureus* bacteremia receiving empirical therapy with cefuroxime, piperacillin-tazobactam, or combination therapy

| Empirical therapy | Adjusted and PS-matched 7-day HR (95 % CI) | Adjusted and PS-matched 30-day HR (95 % CI) | Adjusted and PS-matched 90-day HR (95 % CI) |
|---|---|---|---|
| Cefuroxime monotherapy (n = 237) | 1.00 | 1.00 | 1.00 |
| Piperacillin-tazobactam monotherapy (n = 214) | 0.81 (0.38-1.76) | 0.82 (0.47-1.46) | 0.81 (0.50-1.32) |

examined the effect of exclusive exposure to one of the antimicrobial drugs, which may be misleading as combination EAT is widely used today.

In this study, the primary focus of MS-SAB was only registered if documented by the treating physician in the medical record or verified by a positive *S. aureus* culture from the suspected site of the infection. Otherwise, it was registered as unknown. This may explain the relatively high proportion of cases with an unknown primary focus, ranging from 44.1% to 50.6% in the therapy groups. Of interest, mortality rates in this study are comparable with what has been reported previously (21, 22).

To ensure that prognosis differences based on suspected infection site were considered, the indication of EAT was chosen as one of the covariables in the multivariate and matched analyses. Both the indication of EAT and the primary focus of the infection differed significantly between the therapy groups, which may partly explain the higher mortality in the TZP monotherapy group and the combination therapy group. A higher proportion of cases in these therapy groups had a respiratory infection as the primary focus of MS-SAB, which has previously been associated with higher mortality (23, 24). Besides, patients in the TZP group more often had no specified indication of EAT stated in the health record. One could speculate that the reason why there was no documented indication of EAT in the health record might be because the patient suffered from sepsis. Another possible explanation of this could be that the focus of infection was unknown, which has also been reported to be associated with high mortality (23). Contrary, a larger percentage of the cefuroxime group received therapy on the indication "skin, soft tissue or bone infection". This indication is very heterogeneous, including both patients suspected of having milder infections, such as localized erysipelas, and more serious infections, such as fulminant osteomyelitis. Overall, it seems difficult to conclude whether the indication of EAT may have had an impact on the outcome. Notably, elimination of the indication of EAT from the adjusted analyses only had a marginal impact on the results (unpublished data).

In the study, a higher relative proportion of the patients received cefuroxime as EAT in the earlier years of the study period compared to recent years (Fig. 3), due to changes in empirical antimicrobial guidelines for sepsis from cefuroxime to TZP in the 2010s. We have previously reported that the incidence of MS-SAB increased in Denmark from 2008 to 2015 (25). The result was mainly related to an increased

**TABLE 4** Crude and adjusted hazard ratios of patients with methicillin-susceptible *S. aureus* bacteremia stratified by treatment duration of cefuroxime or piperacillin-tazobactam as empirical monotherapy <3 days or ≥3 days

| Empirical therapy | 7-day mortality | | 30-day mortality | | 90-day mortality | |
|---|---|---|---|---|---|---|
| | Crude HR (95 % CI) | Adjusted HR (95 % CI) | Crude HR (95 % CI) | Adjusted HR (95 % CI) | Crude HR (95 % CI) | Adjusted HR (95 % CI) |
| Empirical monotherapy <3 days (n = 454) | | | | | | |
| Cefuroxime (n = 150) | 1.00 | 1.00 | 1.00 | 1.00 | 1.00 | 1.00 |
| Piperacillin-tazobactam (n = 304) | 0.66 (0.43-1.00) | 0.70 (0.44-1.13) | 0.81 (0.58-1.13) | 0.68 (0.47-0.98) | 0.99 (0.73-1.34) | 0.83 (0.60-1.16) |
| | | | | | | |
| Empirical monotherapy ≥3 days (n = 312) | | | | | | |
| Cefuroxime (n = 187) | 1.00 | 1.00 | 1.00 | 1.00 | 1.00 | 1.00 |
| Piperacillin-tazobactam (n = 125) | 2.98 (1.33-6.68) | 3.77 (1.54-9.24) | 1.79 (1.11-2.89) | 2.04 (1.20-3.46) | 1.70 (1.15-2.49) | 1.78 (1.17-2.72) |

**TABLE 5** Adjusted and propensity score-matched hazard ratios of patients with methicillin-susceptible *S. aureus* bacteremia stratified by treatment duration of cefuroxime or piperacillin-tazobactam as empirical monotherapy <3 days or ≥3 days

| Empirical therapy | Adjusted and PS-matched 7-day HR (95 % CI) | Adjusted and PS-matched 30-day HR (95 % CI) | Adjusted and PS-matched 90-day HR (95 % CI) |
|---|---|---|---|
| Empirical monotherapy <3 days (*n* = 218) | | | |
| Cefuroxime (*n* = 112) | 1.00 | 1.00 | 1.00 |
| Piperacillin-tazobactam (*n* = 106) | 0.87 (0.46-1.64) | 0.76 (0.46-1.25) | 0.87 (0.55-1.37) |
| Empirical monotherapy ≥3 days (*n* = 198) | | | |
| Cefuroxime (*n* =113) | 1.00 | 1.00 | 1.00 |
| Piperacillin-tazobactam (*n* = 85) | 3.79 (1.03-13.88) | 1.70 (0.86-3.38) | 1.66 (0.96-2.86) |

incidence in the older age groups, who also have a higher risk of a fatal outcome of MS-SAB. The increasingly older population in Denmark could potentially mask the effect of TZP compared with cefuroxime as EAT. Nevertheless, no difference in median age was observed between the two monotherapy groups.

Our impression is that the use of diagnostic procedures, including echocardiography, has increased over the last decades. As such, the differences in the proportion of patients who underwent echocardiography in the two treatment groups are likely a consequence of the periods in which the two regimens were predominantly administered (Fig. 3). Moreover, the increasing use of echocardiography in the management of SAB could explain the higher rate of endocarditis in the TZP group compared to the cefuroxime group.

To examine any potential effect of prolonged EAT on the clinical outcome, separate analyses were performed stratified by duration of EAT (<3 days or ≥3 days of treatment). A higher proportion of the cefuroxime monotherapy group received EAT ≥3 days compared to the TZP group. In Denmark, cefuroxime is considered to be sufficient definitive monotherapy for verified MS-SAB. Therefore, one could speculate that the higher proportion of cefuroxime monotherapy in the long EAT duration group is due to a continuation once MS-SAB is verified. TZP alone had a significantly higher crude and adjusted HR of death after 7, 30, and 90 days compared to cefuroxime when administered 3 days or more. However, patients receiving prolonged TZP had significantly more comorbidity compared to patients receiving prolonged cefuroxime, which could explain the higher mortality found for the TZP group (Table S5). This is supported by the fact that after matching covariables, including comorbidity, the risk of mortality was comparable between the two groups. Furthermore, TZP is generally not recommended as a

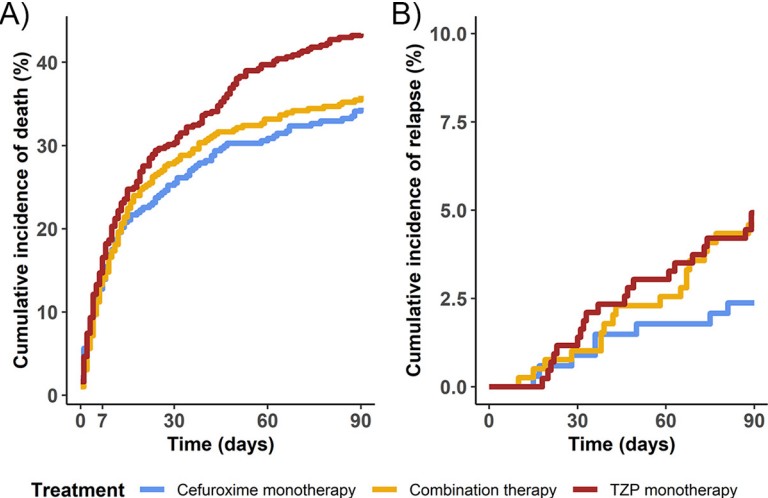

**FIG 2** Cumulative incidence of death (A) or relapse with competing risk of death (B) of methicillin-susceptible *S. aureus* bacteremia based on empirical antimicrobial therapy.

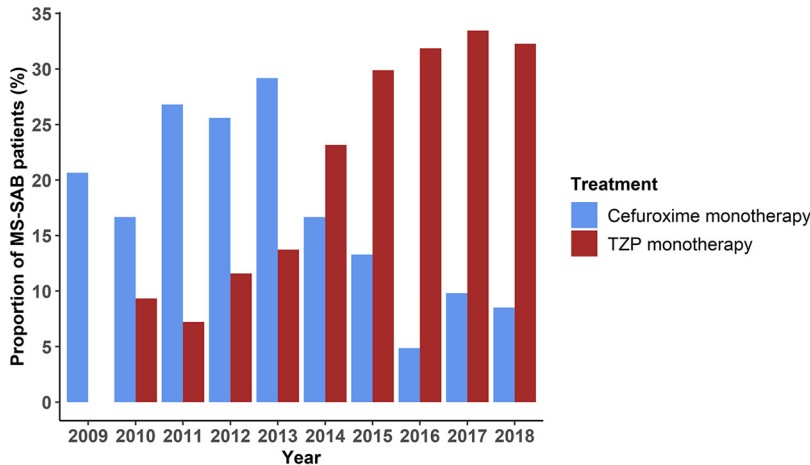

**FIG 3** Proportion of patients with methicillin-susceptible *Staphylococcus aureus* bacteremia who received cefuroxime or piperacillin-tazobactam as empirical monotherapy per calendar year.

definitive treatment for MS-SAB in Denmark, and prolonged treatment with TZP may indicate treatment for more complicated infections such as polymicrobial infection. A potentially higher proportion of patients in the TZP group suffering from more than one infection could influence the clinical outcome of this group. This aligns with a previous report describing higher mortality in MS-SAB patients who were exclusively exposed to TZP compared to patients treated with anti-staphylococcal penicillin or cefazolin (17). Still, further analyses are warranted to conclude on this matter.

No differences in crude or adjusted relapse HR were observed between the therapy groups. Relapse was included as one of the outcomes in this study because it may indicate insufficient antimicrobial therapy, including EAT. Our definition of MS-SAB relapse should cover most relapses caused by insufficient therapy because any relapse after 3 months is unlikely due to only this. Relapse analysis was not performed on the matched population because the matched population was considered too small to get any workable matched relapse results.

The study is not free of limitations. One important limitation is the risk of confounding by indication. As there was no randomization of therapy, the choice of empirical therapy was solely based on the primary assessment of the treating physician and local hospital guidelines. Patients with severe sepsis tend to receive broad-spectrum EAT and often also combination therapy with more than one antimicrobial drug (26, 27). As such, there is a risk that different subgroups of patients were more likely to receive certain EAT regimens above others, which can cause misleading results. Propensity score matching is used in this study to partially adjust for this bias.

The study covers MS-SAB cases over 10 years, and the standard of care for patients with MS-SAB could have changed during this period. We have already described the gradual decrease in the use of cefuroxime and the gradual increase of TZP empirically against MS-SAB in the study period. To the best of our knowledge, no other significant changes in the recommended antimicrobial therapy regimens against MS-SAB were made within the study period.

Dosages of antimicrobial therapy were not registered, and any effects based on different dosages of EAT drugs could therefore not be analyzed. However, most patients received standard dosages of cefuroxime (1.5 g intravenous [IV] × 3) and piperacillin-tazobactam (4 g/0.5 g IV × 3 to 4) per Danish guidelines.

A higher proportion of patients in the cefuroxime group received unspecified definitive therapy, including combination therapy. Consequently, there is a risk that a substantial part of this group received TZP as definitive therapy. However, this seems unlikely because most patients in the cefuroxime group were cases from 2009 to 2013. In this period,

clinical use of TZP against MS-SAB in our region of Denmark was limited compared to today (Fig. 3). A considerable proportion of patients in the TZP group received cefuroxime as definitive therapy, which could potentially impact the results. Contrary, the initial time to appropriate antimicrobial therapy is widely acknowledged as one of the most important factors for MS-SAB and sepsis survival (5, 6, 28). The comparative importance of effective EAT and definitive therapy against MS-SAB is not fully examined (28).

The duration of administered EAT only accounted for a minor part of the total duration of the antimicrobial therapy against MS-SAB in this study. The longer median duration of EAT in the cefuroxime group compared to the TZP group may be caused by the fact that cefuroxime is considered appropriate definitive monotherapy for verified MS-SAB in Denmark. In general, anti-staphylococcal penicillin, penicillin, or cefuroxime is preferred over TZP as definitive therapy for MS-SAB. Consequently, cefuroxime as EAT had a substantially higher probability of continuation compared to TZP once MS-SAB was verified. Additional analyses were performed on the monotherapy groups, in which the effect of EAT duration of 3 days or more was examined. As the time to appropriate antimicrobial therapy has an impact on the outcome of bacteremia, and in particular MS-SAB, it remains crucial to assess whether the effect of specific anti-staphylococcal EAT drugs differs significantly (5, 6, 28).

In conclusion, this study found no significant difference in 7-, 30-, or 90-day mortality or relapse between MS-SAB patients treated with empirical piperacillin-tazobactam or cefuroxime after adjustment and matching of covariables. A randomized clinical trial is warranted to verify the results.

## MATERIALS AND METHODS

**Design and settings.** The study was a retrospective matched cohort study of patients diagnosed with MS-SAB from January 1, 2009 until December 31, 2018.

Patients were identified via the Department of Clinical Microbiology at Amager and Hvidovre Hospital and Statens Serum Institut in Copenhagen, Denmark. The Department of Clinical Microbiology at Amager and Hvidovre Hospital serves the following six hospitals in the greater Copenhagen area: Hvidovre Hospital, Amager Hospital, Glostrup Hospital, Bispebjerg Hospital, Frederiksberg Hospital, and Bornholm Hospital. In 2020, the catchment area included a study population of 1.1 million residents. The study was approved by the Danish Patient Safety Authority (record no. 3-3013-2329/1) and the Regional Data Protection Center (record no. 012-58-0004). Danish legislation does not require informed consent for register-based studies.

**Data sources.** Data were collected by manual review of electronic health records.

**Exclusion criteria.** A case of MS-SAB was defined by the following: (i) at least one positive blood culture of *S. aureus* and (ii) age > 18 years. Cases of MS-SAB suspected to be due to contamination, recurrent MS-SAB within the last 90 days, or polymicrobial bacteremia were excluded at baseline. Cases with polymicrobial bacteremia were excluded because the study solely focused on EAT for MS-SAB.

**Definitions.** Hospital-acquired bacteremia was defined as a positive blood culture of *S. aureus* obtained > 48 h and community-acquired when obtained ≤ 48 h after hospital admission. Healthcare-associated MS-SAB was defined as a case where the patient was in outpatient care (e.g., outpatient dialysis or chemotherapy) or lived at a nursing home. Alcohol abuse, smoking, and immunosuppressive treatment were defined as described previously (19).

**Empirical antimicrobial therapy (EAT).** Appropriate empirical therapy had to meet the following criteria: (i) administration of the antimicrobial drug was initiated before the reporting of the positive blood culture of *S. aureus*, and (ii) the *S. aureus* isolate was susceptible to the relevant antimicrobial drug(s) *in vitro*. EAT given < 24 h was excluded. Three therapy groups were defined: monotherapy with cefuroxime, monotherapy with TZP, and either cefuroxime or TZP in combination with one or more other effective antimicrobials. Cefuroxime and TZP were given intravenously, but effective oral therapy regimens were allowed in the combination therapy group. Cases of MR-SAB were excluded at baseline as both cefuroxime and TZP are considered insufficient antimicrobial therapy regimens for MR-SAB (29).

The indication for empirical therapy at the time of blood culture testing was recorded and classified as one of seven possible indications: (i) "fever with unknown focus", (ii) "skin, soft tissue, or bone infection", (iii) "urinary tract infection", (iv) "IV device infection" (both central and peripheral), (v) "pneumonia", (vi) "not mentioned", and (vii) "other" (e.g., meningitis). Indications for EAT were solely defined by the treating physician who prescribed EAT and only registered if clearly stated in the medical record before blood culture results became available.

Subgroup analyses stratified by duration of EAT (<3 days or ≥3 days of treatment) were performed on the monotherapy groups to examine any potential effect of prolonged EAT on the clinical outcome. Cutoff for the duration of EAT was set to 3 days because results from both blood culture and *in vitro* susceptibility tests would usually be available at this point. Any continuation of EAT after 2 days was therefore considered prolonged EAT.

**Comorbidity score.** Comorbidities at the onset of MS-SAB were registered. Three levels of comorbidities were defined based on the Charlson Comorbidity Index (CCI) score, which has previously been validated for patients with MS-SAB; low comorbidity (CCI = 0), medium comorbidity (CCI = 1 to 2), and high comorbidity (CCI > 2) (30, 31).

**Clinical scores.** SOFA (Sequential Organ Failure Assessment) score and Pitt bacteremia score were calculated for all patients with available data. The clinical scores were based on vital signs and blood test results (only relevant for SOFA score) at MS-SAB onset, defined as the worst combined vital signs within 12 h of blood culture testing and the worst blood test results within 24 h of blood culture testing. If three or more SOFA score parameters were missing, the SOFA score was not calculated. If two or more Pitt score parameters were missing, the Pitt score was not calculated. Pitt bacteremia score ≥ 4 was defined as a critical disease.

**Clinical outcomes.** Outcomes in the study were 7-, 30-, and 90-day all-cause mortality and MS-SAB relapse. MS-SAB relapse was defined as a new episode of MS-SAB or other severe *S. aureus* infection after completion of antimicrobial therapy and < 90 days after the initial MS-SAB episode. A severe *S. aureus* infection was defined by the presence of a secondary manifestation (e.g., endocarditis). If relapse occurred > 90 days after the initial MS-SAB episode, it was regarded as a new, separate episode of MS-SAB.

**Statistics.** Baseline characteristics of the three therapy groups were assessed using chi-square or Fisher exact test for categorical variables and Student's *t* test for continuous variables. All-cause mortality after 7, 30, and 90 days and 90-day relapse were calculated as unadjusted, adjusted, and matched hazard ratios (HR) with a 95% confidence interval (CI) using Cox proportional hazard regression analysis. Odds ratios (OR) with 95% CI were also calculated. The unadjusted mortality differences between all therapy regimens were shown as the cumulative incidence of death. Akaike Information Criterion (AIC) was used to identify valid covariables to avoid overfitting in the adjusted models. The best-fitting model included age, sex, CCI score, the indication of EAT, year of administration, alcohol abuse, smoking, and injection drug use (Table S1). $P < 0.05$ was considered statistically significant. The data analysis program used was R software version 3.6.0 (32).

**Matching.** The monotherapy regimens in this study (cefuroxime and TZP) were matched in the mortality analyses to account for confounding by indication. The matching method used was propensity score matching (PSM). Cases were matched on age, sex, CCI score, an indication of EAT, SOFA score, and Pitt score. The PSM caliper was set to 0.20. Some cases or controls were matched up against more than one case or control if they had more than one single fitting match.

## SUPPLEMENTAL MATERIAL

Supplemental material is available online only.
**SUPPLEMENTAL FILE 1**, PDF file, 0.1 MB.

## ACKNOWLEDGMENTS

The study was partly funded by Hvidovre Hospital's Research Foundation.

The authors have filled out the International Committee of Medical Journal Editors (ICMJE) form and declare no conflict of interest.

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
