## [Reviewer comments · Microbiology Spectrum]

Microbiology Spectrum

Comparable effectiveness of cefuroxime and piperacillin/tazobactam as empirical therapy for methicillin-susceptible *Staphylococcus aureus* bacteremia

Robert Bigseth, Håkon Sandholdt, Andreas Petersen, Christian Andersen, Thomas Benfield, and Louise Thorlacius-Ussing

Corresponding Author(s): Robert Bigseth, Copenhagen University Hospital, Hvidovre

Review Timeline:

Submission Date:	September 11, 2021
Editorial Decision:	December 15, 2021
Revision Received:	March 9, 2022
Accepted:	March 27, 2022

Editor: Bonnie Prokesch

Reviewer(s): The reviewers have opted to remain anonymous.

Transaction Report:

DOI: <https://doi.org/10.1128/spectrum.01530-21>

December 15, 2021

Dr. Robert Strengen Bigseth
Copenhagen University Hospital, Hvidovre
Department of Infectious Diseases
Kettegaard Alle 30
Hvidovre, Capital Region DK-2650
Denmark

Re: Spectrum01530-21 (Comparable effectiveness of cefuroxime and piperacillin/tazobactam as empirical therapy for *Staphylococcus aureus* bacteremia)

Dear Dr. Robert Strengen Bigseth:

Thank you for submitting your manuscript to Microbiology Spectrum. Both reviewers felt that this manuscript would be acceptable for publication with major revisions. Please see the comments in detail below. When submitting the revised version of your paper, please provide (1) point-by-point responses to the issues raised by the reviewers as file type "Response to Reviewers," not in your cover letter, and (2) a PDF file that indicates the changes from the original submission (by highlighting or underlining the changes) as file type "Marked Up Manuscript - For Review Only". Please use this link to submit your revised manuscript - we strongly recommend that you submit your paper within the next 60 days or reach out to me. Detailed instructions on submitting your revised paper are below.

Link Not Available

Sincerely,

Bonnie Prokesch

Journals Department
Reviewer comments:

Reviewer #1 (Comments for the Author):

The manuscript by Bigseth and colleagues represents a multicenter retrospective study aimed to determine the equivalent effectiveness of cefuroxime and piperacillin/tazobactam for the treatment of *Staphylococcus aureus* bacteremia (SAB). The study includes a reasonable size cohort that has the potential to support empiric utilization of piperacillin/tazobactam for the management of sepsis. However, the overall global applicability and interpretation are limited by the comparator (cefuroxime). In addition, several items require clarification to fully interpret the results.

Title:

-The authors should consider clarifying the type of SAB (i.e., PSSA, MSSA) in the title and throughout the manuscript. Clarification is provided in the discussion when illustrating examples of other studies (i.e., MSSA).

Introduction:

-2nd paragraph: The statement that cefuroxime is considered to have a "comparable effect" to other cephalosporins is slightly misleading. The citation provided shows conflicting results in previous studies with a possible trend towards poorer outcomes.

-3rd paragraph: The authors could expand on potentially increased mortality with piperacillin/tazobactam treatment of MSSA to illustrate the need to clarify its role in SAB.

Methods:

-For EAT, was initiation prior to reporting of the culture or initiation prior to culture collection required? Reporting may lag hours to days after culture was collected.

-Were any constraints on the duration of empiric therapy required (e.g., 24 hours)? Or was one dose or more acceptable?

-For the combination group, were there any stipulations to the timing and overlap of the agents?

-Were polymicrobial blood cultures excluded (noted in Figure 1)? I recommend including a discussion of exclusions in the Methods.

-Define mortality in definitions. All-cause mortality?

-How were the indications for empirical defined/collected? How did this differ from the primary focus listed in Table 1? Results allude to how these were defined but not abundantly clear.

-For combination therapy, how was "effective" defined? Is this simply in vitro susceptibility?

-Is there more detail on the propensity scores for each group and how this was addressed in regression analysis?

-The PS matching resulted in significant attrition in the piperacillin/tazobactam group. Would the analysis be better served with comparative inverse probability of treatment weighting analysis?

-Were any of the following confounders collected: source control, septic shock, duration of bacteremia, uncomplicated vs complicated SAB, time to defervescence, and time to effective therapy?

Results:

-A significant amount of patients (~30%) were excluded due to receiving other regimens. What were these regimens? If anti-staph penicillin or cephalosporin, would they have been an additional comparator group?

-3rd sentence: Consider making % out of included cohort instead of the total cohort to provide reader appreciation of division amongst the three regimens for the study cohort.

-Several key differences (e.g., TTE, primary focus, duration of EAT, definitive drug therapy, etc.) could be highlighted in the text.

-No comparison tables are provided for characteristics shown in Table 1 for the PS matched cohorts. This makes it hard to appreciate the effectiveness of the matching. This could be included in supplemental materials.

-Authors provide subgroup characteristics for 3 days or more in Supplementary Figure 3 but not for <3 days. This would provide the reader the opportunity to appreciate if differences existed between the groups.

-A major limitation of the study is that the piperacillin/tazobactam group received a median of 1 day of therapy followed by a majority of patients receiving optimized definitive therapy (i.e., dicloxacillin and penicillin). However, the cefuroxime group received 2.5 days and ~50% were on definitive combination therapy (details not provided). In addition, ~10% of the piperacillin/tazobactam group received cefuroxime therapy as definitive therapy confounding the results. These items should be discussed further in the results and discussion.

Discussion:

-Last paragraph: I would clarify 90-day relapse rate. Also, the same wording is found in the abstract.

-Were there significant changes in the standard of care for the management of SAB in the study countries over the time span other than changes in empiric therapy?

Table 1:

-Were stats not performed on Definitive Therapy Drug?

Tables:

-Notate PS matched in the title where applicable

Reviewer #2 (Comments for the Author):

Summary:

This is a thoughtful multicenter retrospective cohort study that seeks to address the timely issue of empiric antimicrobial therapy, particularly the initiation of a beta-lactam, while balancing risk of mortality from *S aureus* bacteremia.

The study is strengthened by its long study period and large cohort which increases the study's extrinsic validity. It is also enhanced by utilizing propensity matching to address confounding by indication, which strengthens the internal validity of the study. This allows readers to better understand how to contextualize the study's findings and apply them to their own practice.

To strength the paper, I would recommend commenting on the following:

-The study's clarity could be enhanced with the addition of explanations or enhanced precision of language to certain terms such as "blood parameters" or the clinical or research relevance of identifying healthcare-associated SAB as a distinct entity.

-To further succinct and relevant communication of statistical information, consolidating tables such as 2 through five and reviewing how certain supplementary figures/tables or figures contribute to the results of interest. This would allow the addition of data that could not be shown to be presented and contribute further to the study's published findings.

Major essential revisions:

Abstract and Importance

Would state at the start that the study assesses MSSA rather than MRSA to ensure readers are aware of the pathogen at hand; international readers in areas with a predominance of MRSA may suppose MRSA over MSSA at the start.

Methods

Please clarify in the definitions why it was important to define a health-care associated SAB in contrast to hospital and community onset as health-care associated SAB is not brought up later in the paper as an entity with notably differences in clinical outcomes.

Please provide exclusion criteria in this section.

Would clarify whether "iv device infections" included both central line associated infections and peripheral access-associated, or only one of the two.

Please clarify what is meant by "blood parameters" for the Pitt bacteremia score as there are temperature, blood pressure, mechanical ventilation, cardiac arrest, and mental status points but the terminology is not commonly used and could confuse some readers with its lack of specificity.

Please clarify why two clinical scores were utilized and calculated.

Please clarify why 3 days was chosen as the cutoff for duration of therapy.

Results:

Would recommend keeping description of inclusion and exclusion criteria to the methods section for paper organization.

Line 201: Would a table of such findings be possible to present and if not, could the statistical results be summarized here?

Discussion

Line 264: Could the statistical results be presented to quantify what the "marginal impact" was? If so, this should be discussed in the results section.

Would mention how the authors believe these results could be salient for practitioners outside of Denmark, given the international readership of the journal.

Minor essential revisions:

Methods:

Line 121: Would mention Supplemental Table 2 here when referencing effective combination regimens

Line 129: Would consider providing an example of a diagnosis that falls into the "other" category.

Line 146: Please define or provide an example of what is meant by a serious *S. aureus* infection

Line 168: Exactly how many cases or controls were matched up against more than one case or control? For readers who may be unfamiliar with propensity score matching, would consider explaining briefly why multiple matches might have been required.

Results:

Line 181-182: Consider providing a specific odds ratio to quantify how much more likely were the TZP monotherapy group to have a higher CCI. Would also be specific in saying CCI rather than comorbidity itself as the score is a cumulative measure.

Line 183: Please consider providing a specific odds ratio to quantify how much more associated "skin, soft tissue or bone infection" was associated with cefuroxime alone.

Staff Comments:

Preparing Revision Guidelines

Please return the manuscript within 60 days; if you cannot complete the modification within this time period, please contact me. If you do not wish to modify the manuscript and prefer to submit it to another journal, please notify me of your decision immediately so that the manuscript may be formally withdrawn from consideration by Microbiology Spectrum.

Review: Comparable effectiveness of cefuroxime and piperacillin/tazobactam as empirical therapy for *Staphylococcus aureus* bacteremia

Summary:

This is a thoughtful multicenter retrospective cohort study that seeks to address the timely issue of empiric antimicrobial therapy, particularly the initiation of a beta-lactam, while balancing risk of mortality from *S aureus* bacteremia.

The study is strengthened by its long study period and large cohort which increases the study's extrinsic validity. It is also enhanced by utilizing propensity matching to address confounding by indication, which strengthens the internal validity of the study. This allows readers to better understand how to contextualize the study's findings and apply them to their own practice.

To strength the paper, I would recommend commenting on the following:

-The study's clarity could be enhanced with the addition of explanations or enhanced precision of language to certain terms such as "blood parameters" or the clinical or research relevance of identifying healthcare-associated SAB as a distinct entity.

-To further succinct and relevant communication of statistical information, consolidating tables such as 2 through five and reviewing how certain supplementary figures/tables or figures contribute to the results of interest. This would allow the addition of data that could not be shown to be presented and contribute further to the study's published findings.

Major essential revisions:

Abstract and Importance

Would state at the start that the study assesses MSSA rather than MRSA to ensure readers are aware of the pathogen at hand; international readers in areas with a predominance of MRSA may suppose MRSA over MSSA at the start.

Methods

Please clarify in the definitions why it was important to define a health-care associated SAB in contrast to hospital and community onset as health-care associated SAB is not brought up later in the paper as an entity with notably differences in clinical outcomes.

Please provide exclusion criteria in this section.

Would clarify whether "iv device infections" included both central line associated infections and peripheral access-associated, or only one of the two.

Please clarify what is meant by "blood parameters" for the Pitt bacteremia score as there are temperature, blood pressure, mechanical ventilation, cardiac arrest, and mental status points but the terminology is not commonly used and could confuse some readers with its lack of specificity.

Please clarify why two clinical scores were utilized and calculated.

Please clarify why 3 days was chosen as the cutoff for duration of therapy.

Results:

Would recommend keeping description of inclusion and exclusion criteria to the methods section for paper organization.

Line 201: Would a table of such findings be possible to present and if not, could the statistical results be summarized here?

Discussion

Line 264: Could the statistical results be presented to quantify what the "marginal impact" was? If so, this should be discussed in the results section.

Would mention how the authors believe these results could be salient for practitioners outside of Denmark, given the international readership of the journal.

Minor essential revisions:

Methods:

Line 121: Would mention Supplemental Table 2 here when referencing effective combination regimens

Line 129: Would consider providing an example of a diagnosis that falls into the “other” category.

Line 146: Please define or provide an example of what is meant by a serious *S. aureus* infection

Line 168: Exactly how many cases or controls were matched up against more than one case or control? For readers who may be unfamiliar with propensity score matching, would consider explaining briefly why multiple matches might have been required.

Results:

Line 181-182: Consider providing a specific odds ratio to quantify how much more likely were the TZP monotherapy group to have a higher CCI. Would also be specific in saying CCI rather than comorbidity itself as the score is a cumulative measure.

Line 183: Please consider providing a specific odds ratio to quantify how much more associated “skin, soft tissue or bone infection” was associated with cefuroxime alone.

Discretionary revisions: None

Confidential comments: None

Recommendation: Accept with major revisions

Dear editors and reviewers,

Following your letter regarding our manuscript “Comparable effectiveness of cefuroxime and piperacillin/tazobactam as empirical therapy for Staphylococcus aureus bacteremia”, we hereby send this point-by-point response letter which explains the changes performed in the revised manuscript.

We would like to thank the reviewers for taking the time to evaluate our manuscript and for the insightful comments and suggestions. In addition, we would like to thank the editor for the opportunity to resubmit our revised manuscript.

Below is given a point-by-point response to the reviewer comments formatted in italic. Important changes in the revised manuscript are highlighted in yellow.

Reviewer #1

The manuscript by Bigseth and colleagues represents a multicenter retrospective study aimed to determine the equivalent effectiveness of cefuroxime and piperacillin/tazobactam for the treatment of Staphylococcus aureus bacteremia (SAB). The study includes a reasonable size cohort that has the potential to support empiric utilization of piperacillin/tazobactam for the management of sepsis. However, the overall global applicability and interpretation are limited by the comparator (cefuroxime). In addition, several items require clarification to fully interpret the results.

Reviewer comment 1.1

Title:

The authors should consider clarifying the type of SAB (i.e., PSSA, MSSA) in the title and throughout the manuscript.

Clarification is provided in the discussion when illustrating examples of other studies (i.e., MSSA).

Author response 1.1

Thank you for reviewing the manuscript. We agree that it should be stated more clearly that only cases of methicillin-susceptible SAB (MS-SAB) were included in the study. As such, the nomenclature has been changed to MS-SAB throughout the paper, including the title.

Regarding the choice of cefuroxime as the comparator drug to TZP, cefuroxime was chosen because it is the most used cephalosporin against MS-SAB and a prevalent EAT drug in general in Denmark. There has also been a general assertion by clinicians in Denmark that cefuroxime is a better empirical drug against MSSA infections than TZP, which conflicts with the results of our study.

The title has been changed to:

*Comparable effectiveness of cefuroxime and piperacillin/tazobactam as empirical therapy for **methicillin-susceptible** *Staphylococcus aureus* bacteremia*

Reviewer comment 1.2

Introduction:

-2nd paragraph: The statement that cefuroxime is considered to have a "comparable effect" to other cephalosporins is slightly misleading. The citation provided shows conflicting results in previous studies with a possible trend towards poorer outcomes.

Author response 1.2

We agree with the reviewer that prior studies have found somewhat inconsistent results in studies regarding the effect of cephalosporins on the clinical outcome of MS-SAB. Moreover, few studies have specifically assessed different cephalosporins in relation to EAT for SAB. (1–3)

One study that examined the differential effect of empirically administered cephalosporins against MS-SAB found a lower 30-day mortality of cefazolin than cefuroxime and β -lactam- β -lactamase inhibitors (BLBLIs) (incl. TZP). (1) However, this study did not account for confounding by indication and included less than 100 cases in each of the mentioned treatment groups (cefazolin: $n = 28$, cefuroxime: $n = 98$, TZP: $n = 32$. Thus, whether these results would be reproducible with a bigger sample size is unclear. Importantly, a systematic review did not find a differential effect of cephalosporins against MS-SAB. (2) Likewise, a study investigating the inoculum effect of cephalosporins against MSSA did not detect this effect for cefuroxime. (3)

The following has been changed in the manuscript:

Few studies have investigated the effect of cefuroxime compared to other cephalosporins against

MS-SAB and have reported inconsistent findings. (9-11) In Denmark, cefuroxime is the most frequently used cephalosporin against MS-SAB.

Reviewer comment 1.3

-3rd paragraph: The authors could expand on potentially increased mortality with piperacillin/tazobactam treatment of MSSA to illustrate the need to clarify its role in SAB.

Author response 1.3

Thank you for this comment. Prior studies have indicated that TZP is associated with higher 30-day all-cause mortality in patients with MS-SAB. (1,4) However, these studies were based on limited sample sizes and did not properly account for confounding by indication. As such, further investigations are required to conclude on the differential effect of empirical TZP or cefuroxime against MS-SAB.

The following has been changed in the manuscript:

There are, however, some studies that have reported an increased risk of mortality in patients with MS-SAB treated with TZP compared to cephalosporins. (9,17) The sample size of these studies have been limited ($n < 600$), and as such, further investigations are needed to conclude on this matter.

Reviewer comment 1.4

Methods:

-For EAT, was initiation prior to reporting of the culture or initiation prior to culture collection required? Reporting may lag hours to days after culture was collected.

Author response 1.4

Thank you for this comment. All drugs considered for EAT were initiated prior to reporting of blood culture test results. EAT was administered immediately after blood cultures were performed for most SAB patients.

Reviewer comment 1.5

-Were any constraints on the duration of empiric therapy required (e.g., 24 hours)? Or was one dose or more acceptable?

Author response 1.5

Very relevant comment. We only included EAT which was continued for at least 24 hours. This has now been specified in the manuscript.

The following has been changed in the manuscript:

EAT given < 24 hours was excluded.

Reviewer comment 1.6

-For the combination group, were there any stipulations to the timing and overlap of the agents?

Author response 1.6

Thank you for this comment. In general, the combination therapy group consisted of a very heterogeneous group of EAT regimens in terms of timing and duration of overlap of administered EAT drugs. The EAT drugs in the combination therapy group were registered as long as they were initiated before the results of the blood cultures became available. Thus, there were no constraints on whether they had to overlap in time of administration. As an example, if a patient received cefuroxime empirically on day one and two after admission and thereafter dicloxacillin empirically on day two and three before the blood culture was positive of MSSA later on day three, the patient was included in the combination therapy group.

Reviewer comment 1.7

-Were polymicrobial blood cultures excluded (noted in Figure 1)? I recommend including a discussion of exclusions in the Methods.

Author response 1.7

Thank you for this comment. Cases of polymicrobial bloodstream infections were excluded at baseline because this study solely focused on EAT for MS-SAB. An exclusion criteria paragraph with a clarification of polymicrobial bacteremia has been added to the Methods section.

The following has been changed in the manuscript:

Exclusion criteria

A case of MS-SAB was defined by the following: I) at least one positive blood culture of *S. aureus* and II) age > 18 years. Cases of MS-SAB suspected to be due to contamination, recurrent MS-SAB within the last 90 days or polymicrobial bacteremia were excluded at baseline. Cases with polymicrobial bacteremia were excluded because the study solely focused on EAT for MS-SAB.

Reviewer comment 1.8

-Define mortality in definitions. All-cause mortality?

Author response 1.8

Thank you for this comment. All-cause mortality was used in the primary outcome measures of this study, which has now been specified in the manuscript.

The following has been changed in the manuscript:

Clinical outcomes

Outcomes in the study were 7-, 30- and 90-day all-cause mortality and MS-SAB relapse.

Reviewer comment 1.9

-How were the indications for empirical defined/collected? How did this differ from the primary focus listed in Table 1? Results allude to how these were defined but not abundantly clear.

Author response 1.9

Thank you for this very relevant comment. Data on the indication for EAT was only registered if clearly stated in the medical record by the treating physician before blood culture test results became available. The primary focus of the infection was defined as the suspected primary focus of SAB after the blood culture test results were available. As such, the indication for EAT and the primary focus of the infection would likely be similar in most cases but not necessarily.

The following has been changed in the manuscript:

Indications for EAT were solely defined by the treating physician who prescribed EAT and only

registered if clearly stated in the medical record before blood culture results became available.

Reviewer comment 1.10

-For combination therapy, how was "effective" defined? Is this simply in vitro susceptibility?

Author response 1.10

Thank you for this comment. Effective therapy was defined by antimicrobial drugs which the isolate was susceptible in vitro. Thus, for combination therapy, both drugs were required to have in vitro activity. As most antimicrobial therapy regimens are effective against MS-SAB, few cases were excluded due to ineffective EAT (please see figure 1). If a patient received one effective and one ineffective EAT drug, only the effective drug was considered for EAT. Please see the EAT paragraph under Definitions in the Methods section.

Reviewer comment 1.11

-Is there more detail on the propensity scores for each group and how this was addressed in regression analysis?

Author response 1.11

Very relevant comment. We employed propensity score methods, where the predicted probability of treatment with cefuroxime was derived from unconditional logistic regression. The predicted probability of the model was used as the propensity score for each patient. For the propensity score-matched case-control analyses, patients in the cefuroxime therapy group were matched with patients in the TZP therapy group who had the closest propensity scores within a caliper size of one fifth of the standard deviation of the propensity score. There were no restrictions on the ratio of matched cases and controls. In addition, please see author response 2.17.

Reviewer comment 1.12

-The PS matching resulted in significant attrition in the piperacillin/tazobactam group. Would the analysis be better served with comparative inverse probability of treatment weighting analysis?

Author response 1.12

Thank you for this relevant comment. PSM is a well-established method to adjust for confounding

by indication in observational studies. (5) However, as with all other matching or covariable adjustment methods, PSM has both advantages and limitations when used. (6) Inverse probability of treatment weighting analysis (IPTW) differs from PSM primarily on the point that it includes all cases in the analysis. With PSM, outliers are excluded to give proper matching. This could potentially lead to significantly excluded data and in the end affect the analyses and results. However, in our paper, close to 2/3 of all included cases were matched, which we believe is sufficient to be considered representative. Importantly, a prior study that compared PSM and IPTW in four cardiovascular studies showed that PSM performed better than IPTW by providing more precise estimates of treatment effect, especially in cases with a substantial risk of confounding. (6)

Lastly, we do not fully agree that PSM resulted in “significant attrition” in the TZP group, especially if compared to covariable adjustment analysis. As seen in table 2, covariable adjustment analysis had a bigger effect on the hazard ratio (HR) of death after 7, 30 and 90 days in the TZP group compared to PSM.

Reviewer comment 1.13

-Were any of the following confounders collected: source control, septic shock, duration of bacteremia, uncomplicated vs complicated SAB, time to defervescence, and time to effective therapy?

Author response 1.13

Unfortunately, not all of the requested data were available to include in the study.

Control blood cultures are not routinely obtained in patients with SAB in Denmark. As such, data on the duration of bacteremia was not available. Data on source control and time to defervescence were unfortunately not available.

Specific data on septic shock were not collected separately. However, we included data on the Pitt score and SOFA score of SAB cases, which we believe are accurate and informative measures of the severity of the disease and the clinical presentation of SAB.

Regarding complicated and uncomplicated cases of SAB, we registered all patients with any

suspected or verified secondary manifestation. A total of 34.5 % of MS-SAB cases in our study had secondary manifestations of MS-SAB and thus complicated infections, which aligns with prior studies in the field. (7)

Reviewer comment 1.14

Results:

-A significant amount of patients (~30%) were excluded due to receiving other regimens. What were these regimens? If anti-staph penicillin or cephalosporin, would they have been an additional comparator group?

Author response 1.14

Very relevant comment. Several other EAT comparator groups were considered, including dicloxacillin monotherapy, penicillin monotherapy, cefuroxime + gentamycin combination therapy and a group of combination therapy of > 1 effective antimicrobial drug other than cefuroxime + gentamicin. However, multiple EAT comparator groups resulted in such small therapy groups that any statistical comparison to TZP/cefuroxime would be in lack of sufficient statistical power.

Reviewer comment 1.15

-3rd sentence: Consider making % out of included cohort instead of the total cohort to provide reader appreciation of division amongst the three regimens for the study cohort.

Author response 1.15

Excellent suggestion. We have now added these percentages to the manuscript.

The following has been changed in the manuscript:

Of included individuals, 429 (37.0 %) received piperacillin/tazobactam, 337 (29.1 %) received monotherapy with cefuroxime, and 392 (33.9 %) were treated with a combination of cefuroxime or TZP and at least one other anti-staphylococcal drug.

Reviewer comment 1.16

-Several key differences (e.g., TTE, primary focus, duration of EAT, definitive drug therapy, etc.) could be highlighted in the text.

Author response 1.16

We agree with the reviewer that these differences deserve further elaboration in the manuscript.

This has now been added, see below.

Regarding the duration of EAT and definitive drug therapy, please see author response 1.19.

The following has been changed in the manuscript:

Results

Significant differences regarding the primary focus of the infections were observed between the therapy groups. As such, skin and post-operative infections were more frequent in the cefuroxime therapy group. Contrary, the TZP and the combination therapy group had a higher proportion of cases with a primary respiratory/pulmonary focus of the infection compared to the cefuroxime therapy group. The rate of cases with an unknown focus of infection was similar for the two monotherapy groups.

Patients in the TZP and the combination therapy group were more likely to undergo echocardiography, both transthoracic (TTE) and transesophageal (TEE), compared to the cefuroxime group.

Discussion

Our impression is that the use of diagnostic procedures, including echocardiography, has increased over the last decades. As such, the differences in the proportion of patients who underwent echocardiography in the two treatment groups are likely a consequence of the time periods in which the two regimens were predominantly administered (Figure 3). Moreover, the increasing use of echocardiography in the management of SAB could explain the higher rate of endocarditis in the TZP group compared to the cefuroxime group.

Reviewer comment 1.17

-No comparison tables are provided for characteristics shown in Table 1 for the PS matched cohorts. This makes it hard to appreciate the effectiveness of the matching. This could be included in supplemental materials.

Author response 1.17

We agree with the reviewer that more details on the characteristics of the propensity score-matched cohort would add significant information on the effectiveness of the matching process. As such, a table displaying baseline characteristics of matched cases has been provided in Supplementary Material.

The following has been changed in the manuscript:

Supplementary Table 4: Characteristics of the propensity score-matched *Staphylococcus aureus* bacteremia cases stratified by empirical monotherapy with cefuroxime or piperacillin/tazobactam

	Cefuroxime monotherapy (n = 237)	Piperacillin/tazobactam monotherapy (n = 214)	P-value
Female (%)	93 (39.2)	82 (38.3)	0.92
Median age (IQR)	74 (62-84)	74 (61-84)	0.81
Comorbidity score (CCI)			
Low, CCI = 0 (%)	52 (21.9)	32 (15.0)	
Medium, CCI = 1-2 (%)	88 (37.1)	81 (37.9)	
High, CCI > 2 (%)	97 (40.9)	101 (47.2)	0.14
Median SOFA score at onset (IQR)	2 (1-4)	3 (1-4)	0.92
Pitt score ≥ 4 at onset (%)	18 (7.6)	11 (5.1)	0.38
Indication of EAT			
Fever with unknown focus (%)	58 (24.5)	66 (30.8)	
Skin, soft tissue or bone infection (%)	45 (19.0)	25 (11.7)	
Urinary tract infection (%)	24 (10.1)	27 (12.6)	
Iv device infection (%)	7 (3.0)	2 (0.9)	
Pneumonia (%)	61 (25.7)	56 (26.2)	
Not mentioned (%)	23 (9.7)	28 (13.1)	
Other (%)	19 (8.0)	10 (4.7)	0.073

Reviewer comment 1.18

-Authors provide subgroup characteristics for 3 days or more in Supplementary Figure 3 but not for <3 days. This would provide the reader the opportunity to appreciate if differences existed between the groups.

Author response 1.18

Thank you for this relevant comment. A table including information on patients receiving less than three days of EAT with cefuroxime or TZP has been added to the manuscript.

The following has been changed in the manuscript:

Supplementary Table 5: Characteristics of *Staphylococcus aureus* bacteremia cases stratified by

empirical monotherapy with cefuroxime or piperacillin/tazobactam administered (A) less than three days (B) or three days or more

(A)

	Cefuroxime monotherapy < three days (n = 150)	Piperacillin/ tazobactam monotherapy < three days (n = 304)	P-value
Female (%)	60 (40.0)	119 (39.1)	0.94
Median age (IQR)	73.5 (59-82)	75 (62-84)	0.35
Comorbidity score (CCI)			
Low, CCI = 0 (%)	36 (24.0)	47 (15.5)	
Medium, CCI = 1-2 (%)	50 (33.3)	124 (40.8)	
High, CCI > 2 (%)	64 (42.7)	133 (43.8)	0.064
Smoking (%)	33 (22.0)	66 (21.7)	0.99
Daily alcohol consumption (%)	28 (18.7)	75 (24.5)	0.23
Injection drug use (%)	7 (4.7)	11 (3.6)	0.97
Any immunosuppression* (%)	7 (4.7)	18 (5.9)	0.69
Median SOFA score at onset (IQR)	3 (1-4)	3 (1-4)	0.72
Pitt score ≥ 4 at onset (%)	13 (10.3)	14 (5.4)	0.12
Secondary manifestations			
Endocarditis (%)	7 (4.7)	35 (11.5)	0.028
Osteomyelitis (%)	10 (6.7)	12 (3.9)	0.30
Spondylodiscitis (%)	11 (7.3)	18 (5.9)	0.71
Arthritis (%)	9 (6.0)	14 (4.6)	0.68
Meningitis (%)	0 (0.0)	2 (0.7)	0.81
Pneumonia (%)	11 (7.3)	20 (6.6)	0.92
Other (%)	9 (6.0)	10 (3.3)	0.27
Relapse within 90 days (%)	4 (2.7)	11 (3.6)	0.80
7-day mortality (%)	38 (25.3)	54 (17.8)	0.078
30-day mortality (%)	54 (36.0)	95 (31.2)	0.36
90-day mortality (%)	63 (42.0)	134 (44.1)	0.75

Reviewer comment 1.19

-A major limitation of the study is that the piperacillin/tazobactam group received a median of 1 day of therapy followed by a majority of patients receiving optimized definitive therapy (i.e., dicloxacillin and penicillin). However, the cefuroxime group received 2.5 days and ~50% were on definitive combination therapy (details not provided). In addition, ~10% of the piperacillin/tazobactam group received cefuroxime therapy as definitive therapy confounding the results. These items should be discussed further in the results and discussion.

Author response 1.19

Thank you for this insightful comment. We agree that the difference in the median duration of EAT in the two therapy groups and the risk of potential confounding as of patients in the TZP group receiving cefuroxime as definitive therapy and vice versa need to be further addressed in the manuscript.

The difference in the median duration of EAT is likely explained by the fact that cefuroxime is considered appropriate definitive monotherapy for MS-SAB in Denmark. As such, once MS-SAB is microbiologically confirmed, TZP is more likely to be changed to another definitive antimicrobial therapy drug compared to cefuroxime. Importantly, multiple studies have shown that time to effective empirical therapy is one of the most important factors related to sepsis survival. (8–11) To the best of our knowledge, the evidence on the duration of EAT and clinical outcomes related to bacteremia and sepsis is limited. As such, early initiation of empirical TZP potentially had a substantial effect on clinical outcomes even though the median duration of EAT was shorter than for cefuroxime.

The following has been changed in the manuscript:

Results

The median duration of EAT was significantly longer in the cefuroxime monotherapy group (2.5 days, IQR 1-6) compared to the TZP monotherapy group (1 day, IQR 1-2) and the combination therapy group (1.5 days, IQR 1-3). Also, the choice of definitive antimicrobial therapy differed between the therapy groups. A higher proportion of patients in the cefuroxime group (50.1 %) received unspecified definitive antimicrobial therapy including combination therapy compared to the other groups (TZP: 22.1 %, comb. therapy: 43.1 %). In the TZP group, 9.8 % of the patients received cefuroxime as definitive therapy.

Discussion

A higher proportion of patients in the cefuroxime group received unspecified definitive therapy, including combination therapy. Consequently, there is a risk that a substantial part of this group received TZP as definitive therapy. However, this seems unlikely, as the majority of patients in the cefuroxime group were cases from 2009 to 2013. In this period, clinical use of TZP against MS-SAB in our region of Denmark was limited compared to today (Figure 3). A considerable proportion of patients in the TZP group received cefuroxime as definitive therapy, which could potentially impact the results. Contrary, the initial time to appropriate antimicrobial therapy is widely acknowledged as one of the most important factors for MS-SAB and sepsis survival. (5,6,32) The comparative importance of effective EAT and definitive therapy against MS-SAB is not fully examined. (32)

The longer median duration of EAT in the cefuroxime group compared to the TZP group may be caused by the fact that cefuroxime is considered appropriate definitive monotherapy for verified MS-SAB in Denmark. In general, anti-staphylococcal penicillin, penicillin or cefuroxime is preferred over TZP as definitive therapy for MS-SAB. Consequently, cefuroxime as EAT had a substantially higher probability of continuation compared to TZP once MS-SAB was verified.

Reviewer comment 1.20

Discussion:

-Last paragraph: I would clarify 90-day relapse rate. Also, the same wording is found in the abstract.

Author response 1.20

We agree with the reviewer that the word “rate” may be unnecessary and has consequently been removed from the paper.

The following has been changed in the manuscript:

Conclusion: *There was no significant difference in 7-, 30- or 90-day mortality or relapse **rate** between MS-SAB patients treated with empirical TZP or cefuroxime after adjustment and matching of covariables.*

*In conclusion, this study found no significant difference in 7-, 30- or 90-day mortality or relapse **rate** between MS-SAB patients treated with empirical piperacillin/tazobactam or cefuroxime after adjustment and matching of covariables.*

Reviewer comment 1.21

-Were there significant changes in the standard of care for the management of SAB in the study countries over the time span other than changes in empiric therapy?

Author response 1.21

Data was collected from six hospitals in the greater Copenhagen area of Denmark, including academic and non-academic hospitals. Although there have been no official guideline changes in the management of patients with SAB, it can not be precluded entirely that diagnostic procedures

may have improved during the study period (e.g. increasing use of echocardiography) which should be acknowledged as a possible limitation of the study.

The following has been changed in the manuscript:

The study covers MS-SAB cases over a ten year period of time. It is possible that the standard of care for patients with MS-SAB could have changed during this period. We have already described the gradual decrease in the use of cefuroxime and the gradual increase of TZP empirically against MS-SAB in the study period. To the best of our knowledge, no other significant changes in the recommended antimicrobial therapy regimens against MS-SAB were made within the study period.

Reviewer comment 1.22

Table 1:

-Were stats not performed on Definitive Therapy Drug?

Author response 1.22

Thank you for this question Although highly relevant, analyses on definitive therapy were not performed as this study specifically addressed the effect of EAT and thus evaluation of definitive therapy on the clinical outcome of SAB is beyond the scope of the present paper.

Reviewer comment 1.23

Tables:

-Notate PS matched in the title where applicable

Author response 1.23

We agree with the reviewer that it would be appropriate to include “propensity score-matched” in the title of relevant tables.

The following has been changed in the manuscript:

Table 3 (A) Crude and adjusted and (B) adjusted and propensity score-matched hazard ratios of patients with methicillin-susceptible *S. aureus* bacteremia stratified by treatment duration of cefuroxime or piperacillin/tazobactam as empirical monotherapy < three days or \geq three days
(B)

	Adjusted and PS-matched 7-day HR (95 % CI)	Adjusted and PS-matched 30-day HR (95 % CI)	Adjusted and PS-matched 90-day HR (95 % CI)
--	---	--	--

Reviewer #2

Summary:

This is a thoughtful multicenter retrospective cohort study that seeks to address the timely issue of empiric antimicrobial therapy, particularly the initiation of a beta-lactam, while balancing risk of mortality from *S aureus* bacteremia.

The study is strengthened by its long study period and large cohort which increases the study's extrinsic validity. It is also enhanced by utilizing propensity matching to address confounding by indication, which strengthens the internal validity of the study. This allows readers to better understand how to contextualize the study's findings and apply them to their own practice.

Reviewer comment 2.1

To strength the paper, I would recommend commenting on the following:

-The study's clarity could be enhanced with the addition of explanations or enhanced precision of language to certain terms such as "blood parameters" or the clinical or research relevance of identifying healthcare-associated SAB as a distinct entity.

Author response 2.1

Thank you for reviewing our manuscript and for the very useful comments.

Regarding precision of language, please see author response 2.7.

Regarding healthcare-associated SAB, please see author response 2.4.

Reviewer comment 2.2

-To further succinct and relevant communication of statistical information, consolidating tables such as 2 through five and reviewing how certain supplementary figures/tables or figures contribute to the results of interest. This would allow the addition of data that could not be shown to be presented and contribute further to the study's published findings.

Author response 2.2

Excellent suggestion. We agree that consolidating the five tables included in the manuscript could enhance the overview of study data. Tables 2 and 3 and tables 4 and 5, respectively, show different data on the same cohorts. Therefore, tables 2 and 3 and tables 4 and 5 have been merged into an A and B section.

The following has been changed in the manuscript:

Table 2: (A) Crude and adjusted and (B) adjusted and propensity score-matched hazard ratios of patients with methicillin-susceptible *S. aureus* bacteremia receiving empirical therapy with cefuroxime, piperacillin/tazobactam or combination therapy

(A)

(B)

Table 3: (A) Crude and adjusted and (B) adjusted and propensity score-matched hazard ratios of patients with methicillin-susceptible *S. aureus* bacteremia stratified by treatment duration of cefuroxime or piperacillin/tazobactam as empirical monotherapy < three days or \geq three days

(A)

(B)

Reviewer comment 2.3

Major essential revisions:

Abstract and Importance

-Would state at the start that the study assesses MSSA rather than MRSA to ensure readers are aware of the pathogen at hand; international readers in areas with a predominance of MRSA may suppose MRSA over MSSA at the start.

Author response 2.3

Thank you for this comment. Please see author response 1.1.

Reviewer comment 2.4

Methods

-Please clarify in the definitions why it was important to define a health-care associated SAB in contrast to hospital and community onset as health-care associated SAB is not brought up later in the paper as an entity with notably differences in clinical outcomes.

Author response 2.4

Thank you for this comment. Healthcare-associated acquisition has been recognized as a separate acquisition group in prior studies on bacteremia and in particular SAB. (1,12–15). Patients with healthcare-associated SAB are often residences at specialized nursing homes or have regular outpatient contact. (12) As such, these patients may represent a specific subgroup of fragile patients with SAB. Moreover, prior studies have stated that patients with healthcare-associated SAB differ on comorbidities, the primary focus of the infection and the risk of a fatal outcome of SAB when compared to patients with community-acquired bacteremia. (12). Any potential difference in the proportion of patients with healthcare-associated MS-SAB between the treatment groups could potentially affect the results of the study. However, this was not the case in this study, as the distribution of mode of onset of SAB was comparable between treatment groups.

Reviewer comment 2.5

-Please provide exclusion criteria in this section.

Author response 2.5

We agree with the reviewer that exclusion criteria should be stated clearly in the Method section. Exclusion criteria in the study comprised the following: cases suspected to be due to contamination rather than infection, cases with recurrent SAB within 90 days of the index date and cases with polymicrobial bacteremia.

The following has been changed in the manuscript:

Exclusion criteria

*A case of MS-SAB was defined by the following: I) at least one positive blood culture of *S. aureus* and II) age > 18 years. Cases of MS-SAB suspected to be due to contamination, recurrent MS-SAB within the last 90 days or polymicrobial bacteremia were excluded at baseline. Cases with polymicrobial bacteremia were excluded because the study solely focused on EAT for MS-SAB.*

Reviewer comment 2.6

-Would clarify whether "iv device infections" included both central line associated infections and peripheral access-associated, or only one of the two.

Author response 2.6

We agree with the reviewer that further clarification on the definition of iv device infections would be appropriate. For this study, iv device infections included both central and peripheral line-associated infections.

The following has been changed in the manuscript:

... 4) "iv device infection" **(both central and peripheral)**...

Reviewer comment 2.7

-Please clarify what is meant by "blood parameters" for the Pitt bacteremia score as there are temperature, blood pressure, mechanical ventilation, cardiac arrest, and mental status points but the terminology is not commonly used and could confuse some readers with its lack of specificity.

Author response 2.7

Thank you for this comment. "Blood parameters" refers to blood test results, which were solely included in the SOFA score and not in the Pitt score. We agree with the reviewer that this could be specified in the manuscript.

The following has been changed in the manuscript:

*The clinical scores were based on vital signs and blood **test results** **(only relevant for SOFA score)** at MS-SAB onset, defined as the worst combined vital signs within 12 hours of blood culture testing and the worst blood **test results** within 24 hours of blood culture testing.*

Reviewer comment 2.8

-Please clarify why two clinical scores were utilized and calculated.

Author response 2.8

We believe that the inclusion of two well-established clinical scores offers a comprehensive

measure of the clinical severity of SAB. In a large study on community-onset bacteremia, the Pitt score has been shown to predict 28-day mortality. (16) The SOFA score is a widely-used tool to quantify the severity of sepsis.

Reviewer comment 2.9

-Please clarify why 3 days was chosen as the cutoff for duration of therapy.

Author response 2.9

Very relevant comment. The cutoff of three days for EAT was chosen because both blood culture and in vitro susceptibility test results would usually be available after this time point. Thus, we believe that three days is an appropriate duration of time when evaluating EAT.

The following has been changed in the manuscript:

Cutoff for the duration of EAT was set to three days because results from both blood culture and in vitro susceptibility tests would usually be available at this point. Any continuation of EAT after two days was therefore considered as prolonged EAT.

Reviewer comment 2.10

Results:

-Would recommend keeping description of inclusion and exclusion criteria to the methods section for paper organization.

Author response 2.10

Thank you for this comment. Description of inclusion and exclusion criteria has now been removed from the Results section.

The following has been changed in the manuscript:

Of 1969 cases of MS-SAB, 168 (8.5 %) did not receive any EAT, 21 (1.1 %) received ineffective EAT, and 622 (31.6 %) received other regimens than those studied here (Figure 1).

Reviewer comment 2.11

-Line 201: Would a table of such findings be possible to present and if not, could the statistical results be summarized here?

Author response 2.11

Thank you for this question. These data were removed as part of a filtering process to shorten the paper. However, we agree with the reviewer that presentation of these data may be of interest for the reader, and thus have provided a table presenting results on separate combination therapy groups in the Supplementary Material.

The following has been changed in the manuscript:

Analyses on separate combination therapy groups including either TZP or cefuroxime was performed and did not show any significant differences in 7-, 30- or 90-day mortality or relapse in crude or adjusted analyses compared to cefuroxime monotherapy (Supplementary Table 3).

Supplementary Table 3: Crude and adjusted hazard ratios of 7-, 30- or 90-day mortality or 90-day relapse for patients with methicillin-susceptible *S. aureus* bacteremia receiving empirical therapy with cefuroxime monotherapy, piperacillin/tazobactam monotherapy or combination therapy with either cefuroxime or piperacillin/tazobactam

	7-day mortality		30-day mortality		90-day mortality		90-day relapse	
	Crude HR (95 % CI)	Adjusted HR (95 % CI)	Crude HR (95 % CI)	Adjusted HR (95 % CI)	Crude HR (95 % CI)	Adjusted HR (95 % CI)	Crude HR (95 % CI)	Adjusted HR (95 % CI)
Cefuroxime monotherapy (n = 337)	1.00	1.00	1.00	1.00	1.00	1.00	1.00	1.00
Piperacillin/tazobactam monotherapy (n = 429)	1.19 (0.83-1.73)	1.03 (0.60-1.75)	1.22 (0.93-1.60)	1.09 (0.71-1.66)	1.33 (1.06-1.68)	0.98 (0.71-1.37)	2.08 (0.9-4.71)	1.51 (0.52-4.33)
Cefuroxime comb. therapy (n = 195)	0.83 (0.50-1.37)	0.96 (0.52-1.74)	0.92 (0.65-1.32)	1.13 (0.69-1.84)	0.91 (0.66-1.23)	1.10 (0.75-1.62)	2.17 (0.86-5.50)	1.97 (0.62-6.32)
Piperacillin/tazobactam comb. therapy (n = 197)	1.16 (0.74-1.81)	1.10 (0.59-2.06)	1.28 (0.93-1.77)	1.08 (0.67-1.75)	1.23 (0.92-1.64)	1.14 (0.77-1.69)	1.73 (0.65-4.61)	1.48 (0.42-5.18)

Reviewer comment 2.12

Discussion

-Line 264: Could the statistical results be presented to quantify what the "marginal impact" was? If so, this should be discussed in the results section.

Author response 2.12

Very relevant comment. Generally, we agree with the reviewer that any referred data should be presented in the paper. However, in this specific example, the requested analyses on HR of death excluding indication of EAT as a covariable were almost identical to the already presented analyses on adjusted HR of death including indication of EAT (see Supplementary Table X below).

Therefore, we decided to exclude this table from the paper. We kindly hope this can be accepted.

Supplementary Table X: *Adjusted hazard ratios with and without indication of empirical therapy as covariable of patients with methicillin-susceptible S. aureus bacteremia receiving empirical therapy with cefuroxime, piperacillin/tazobactam or combination therapy*

	7-day mortality		30-day mortality		90-day mortality	
	Adjusted HR (95 % CI) including indication	Adjusted HR (95 % CI) excluding indication	Adjusted HR (95 % CI) including indication	Adjusted HR (95 % CI) excluding indication	Adjusted HR (95 % CI) including indication	Adjusted HR (95 % CI) excluding indication
Cefuroxime monotherapy (n = 337)	1.00	1.00	1.00	1.00	1.00	1.00
Piperacillin/ tazobactam monotherapy (n = 429)	1.03 (0.60- 1.75)	0.99 (0.59- 1.68)	1.09 (0.71- 1.66)	1.06 (0.70-1. 60)	0.98 (0.71- 1.37)	0.99 (0.71- 1.37)
Combination therapy (n = 392)	1.02 (0.62- 1.67)	0.93 (0.57- 1.51)	1.10 (0.74- 1.64)	1.06 (0.71-1. 56)	1.12 (0.82- 1.53)	1.09 (0.80- 1.48)

Reviewer comment 2.13

-Would mention how the authors believe these results could be salient for practitioners outside of Denmark, given the international readership of the journal.

Author response 2.13

Thank you for this comment. There are multiple reasons why these results are important to practitioners outside Denmark. MS-SAB is still the predominant cause of SAB worldwide, and contemporary data show that the incidence of methicillin-resistant SAB (MR-SAB) is decreasing. (17,18) As MS-SAB is a serious infection with a high risk of fatal outcome, we need sufficient evidence on EAT to optimize antimicrobial therapy and clinical outcomes against this condition.

Regarding the choice of cefuroxime as the comparator drug to TZP related to international readership, please see author response 1.1.

The following has been changed in the manuscript:

Methicillin-susceptible Staphylococcus aureus bacteremia (MS-SAB) is one of the foremost causes of Gram-positive bacteremia and therefore an important cause of sepsis. (1) Although the incidence of methicillin-resistant SAB (MR-SAB) increased during the last decade, more contemporary data have found decreasing rates of MR-SAB and MS-SAB is still the predominant cause of SAB in many western countries. (2-4)

Reviewer comment 2.14

Minor essential revisions:

Methods:

-Line 121: Would mention Supplemental Table 2 here when referencing effective combination regimens

Author response 2.14

Thank you for this comment. Due to paper organization, we prefer not to refer to Supplementary Table 2 in the method section, as this would cause results to be presented in this section. We kindly hope this can be accepted.

Reviewer comment 2.15

-Line 129: Would consider providing an example of a diagnosis that falls into the "other" category.

Author response 2.15

Thank you for this comment. The group of "other" indications for EAT consisted of a wide variety

of clinical manifestations such as meningitis, CNS/lung abscess or endocarditis.

The following has been changed in the manuscript:

... and 7) “other” (e.g. meningitis).

Reviewer comment 2.16

-Line 146: Please define or provide an example of what is meant by a serious *S. aureus* infection

Author response 2.16

We agree with the reviewer that the term “serious“ *S. aureus* infection may be confusing and has now been changed into “severe” *S. aureus* infection, which was defined by the presence of a secondary manifestation such as osteomyelitis, arthritis, endocarditis, pneumonia or meningitis.

The following has been changed in the manuscript:

MS-SAB relapse was defined as a new episode of MS-SAB or other **severe** *S. aureus* infection after completion of antimicrobial therapy and < 90 days after the initial MS-SAB episode. **A severe *S. aureus* infection was defined by the presence of a secondary manifestation (e.g. endocarditis).**

Reviewer comment 2.17

-Line 168: Exactly how many cases or controls were matched up against more than one case or control? For readers who may be unfamiliar with propensity score matching, would consider explaining briefly why multiple matches might have been required.

Author response 2.17

Thank you for this comment. Below is a table showing the exact ratio of matched cases and controls after propensity score matching. Studies have shown that matching with caliper restriction but with no restriction on the ratio of matched cases per control (as done in this paper) enhances the matching quality. (19,20) In addition, please see author response 1.12.

Ratio	n
1:1	195
2:1	17

3:1	0
4:1	2

Reviewer comment 2.18

Results:

-Line 181-182: Consider providing a specific odds ratio to quantify how much more likely were the TZP monotherapy group to have a higher CCI. Would also be specific in saying CCI rather than comorbidity itself as the score is a cumulative measure.

Author response 2.18

Thank you for this comment. We used the Charlson comorbidity index score as a total measure of the burden of comorbidity of the study population. We agree with the reviewer that it would be more accurate to state that the TZP group had a higher CCI score compared to the other therapy groups. An odds ratio of this relation has been added to the paper in order to quantify the higher CCI for the TZP group.

The following has been changed in the manuscript:

Cases in the TZP monotherapy group were more likely to have a higher CCI score compared to the other groups (OR 1.88 [95 % CI 1.30-2.73] for TZP compared to cefuroxime).

Reviewer comment 2.19

-Line 183: Please consider providing a specific odds ratio to quantify how much more associated "skin, soft tissue or bone infection" was associated with cefuroxime alone.

Author response 2.19

Thank you for this relevant comment. We agree with the reviewer that a specific odds ratio to quantify the association between the indication "skin, soft tissue or bone infection" and cefuroxime compared to TZP is relevant.

The following has been changed in the manuscript:

"Skin, soft tissue or bone infection" was more often associated with administration of cefuroxime alone indication (OR 2.79 [95 % CI 1.80-4.32] for cefuroxime compared to TZP)...

References

1. Paul M, Zemer-Wassercug N, Talker O, Lishtzinsky Y, Lev B, Samra Z, et al. Are all beta-lactams similarly effective in the treatment of methicillin-sensitive *Staphylococcus aureus* bacteraemia? *Clin Microbiol Infect*. 2011 Oct;17(10):1581–6.
2. Vardakas KZ, Apiranthiti KN, Falagas ME. Antistaphylococcal penicillins versus cephalosporins for definitive treatment of methicillin-susceptible *Staphylococcus aureus* bacteraemia: a systematic review and meta-analysis. *Int J Antimicrob Agents*. 2014 Dec;44(6):486–92.
3. Nannini EC, Stryjewski ME, Singh KV, Rude TH, Corey GR, Fowler VG, et al. Determination of an Inoculum Effect with Various Cephalosporins among Clinical Isolates of Methicillin-Susceptible *Staphylococcus aureus*. *Antimicrob Agents Chemother*. 2010 May;54(5):2206–8.
4. Beganovic M, Cusumano JA, Lopes V, LaPlante KL, Caffrey AR. Comparative Effectiveness of Exclusive Exposure to Nafcillin or Oxacillin, Cefazolin, Piperacillin/Tazobactam, and Fluoroquinolones Among a National Cohort of Veterans With Methicillin-Susceptible *Staphylococcus aureus* Bloodstream Infection. *Open Forum Infect Dis*. 2019 Jul;6(7):ofz270.
5. Austin PC, Stuart EA. The performance of inverse probability of treatment weighting and full matching on the propensity score in the presence of model misspecification when estimating the effect of treatment on survival outcomes. *Stat Methods Med Res*. 2017 Aug;26(4):1654–70.
6. Elze MC, Gregson J, Baber U, Williamson E, Sartori S, Mehran R, et al. Comparison of Propensity Score Methods and Covariate Adjustment: Evaluation in 4 Cardiovascular Studies. *Journal of the American College of Cardiology*. 2017 Jan 24;69(3):345–57.
7. Horino T, Hori S. Metastatic infection during *Staphylococcus aureus* bacteremia. *Journal of Infection and Chemotherapy*. 2020 Feb 1;26(2):162–9.
8. Kadri SS, Lai YL, Warner S, Strich JR, Babiker A, Ricotta EE, et al. Inappropriate empirical antibiotic therapy for bloodstream infections based on discordant in-vitro susceptibilities: a retrospective cohort analysis of prevalence, predictors, and mortality risk in US hospitals. *Lancet Infect Dis*. 2021 Feb;21(2):241–51.
9. Niederman MS, Baron RM, Bouadma L, Calandra T, Daneman N, DeWaele J, et al. Initial antimicrobial management of sepsis. *Crit Care*. 2021 Aug 26;25(1):307.
10. Ferrer R, Martin-Loeches I, Phillips G, Osborn TM, Townsend S, Dellinger RP, et al. Empiric Antibiotic Treatment Reduces Mortality in Severe Sepsis and Septic Shock From the First Hour: Results From a Guideline-Based Performance Improvement Program*. *Critical Care Medicine*. 2014 Aug;42(8):1749–1755.
11. Buckman SA, Turnbull IR, Mazuski JE. Empiric Antibiotics for Sepsis. *Surg Infect (Larchmt)*. 2018 Mar;19(2):147–54.
12. Friedman ND, Kaye KS, Stout JE, McGarry SA, Trivette SL, Briggs JP, et al. Health care--associated bloodstream infections in adults: a reason to change the accepted definition of community-acquired infections. *Ann Intern Med*. 2002 Nov 19;137(10):791–7.

13. Thønnings S, Jansåker F, Gradel KO, Styrihave B, Knudsen JD. Cefuroxime compared to piperacillin/tazobactam as empirical treatment of *Escherichia coli* bacteremia in a low Extended-spectrum beta-lactamase (ESBL) prevalence cohort. *Infect Drug Resist.* 2019 May 13;12:1257–64.
14. Rasmussen JB, Knudsen JD, Arpi M, Schönheyder HC, Benfield T, Ostergaard C. Relative efficacy of cefuroxime versus dicloxacillin as definitive antimicrobial therapy in methicillin-susceptible *Staphylococcus aureus* bacteraemia: a propensity-score adjusted retrospective cohort study. *J Antimicrob Chemother.* 2014 Feb;69(2):506–14.
15. Forsblom E, Ruotsalainen E, Järvinen A. Comparable Effectiveness of First Week Treatment with Anti-Staphylococcal Penicillin versus Cephalosporin in Methicillin-Sensitive *Staphylococcus aureus* Bacteremia: A Propensity-Score Adjusted Retrospective Study. *PLoS ONE.* 2016;11(11):e0167112.
16. Lee C-C, Lee C-H, Hong M-Y, Tang H-J, Ko W-C. Timing of appropriate empirical antimicrobial administration and outcome of adults with community-onset bacteremia. *Crit Care.* 2017 May 26;21(1):119.
17. Monaco M, Pimentel de Araujo F, Cruciani M, Coccia EM, Pantosti A. Worldwide Epidemiology and Antibiotic Resistance of *Staphylococcus aureus*. In: Bagnoli F, Rappuoli R, Grandi G, editors. *Staphylococcus aureus: Microbiology, Pathology, Immunology, Therapy and Prophylaxis* [Internet]. Cham: Springer International Publishing; 2017 [cited 2022 Jan 16]. p. 21–56. (Current Topics in Microbiology and Immunology). Available from: https://doi.org/10.1007/82_2016_3
18. Tong SYC, Davis JS, Eichenberger E, Holland TL, Fowler VG. *Staphylococcus aureus* infections: epidemiology, pathophysiology, clinical manifestations, and management. *Clin Microbiol Rev.* 2015 Jul;28(3):603–61.
19. Austin PC, Stuart EA. The effect of a constraint on the maximum number of controls matched to each treated subject on the performance of full matching on the propensity score when estimating risk differences. *Stat Med.* 2021 Jan 15;40(1):101–18.
20. Austin PC, Stuart EA. Estimating the effect of treatment on binary outcomes using full matching on the propensity score. *Stat Methods Med Res.* 2017 Desember;26(6):2505–25.

March 27, 2022

Dr. Robert Strengen Bigseth
Copenhagen University Hospital, Hvidovre
Department of Infectious Diseases
Kettegaard Alle 30
Hvidovre, Capital Region DK-2650
Denmark

Re: Spectrum01530-21R1 (Comparable effectiveness of cefuroxime and piperacillin/tazobactam as empirical therapy for methicillin-susceptible *Staphylococcus aureus* bacteremia)

Dear Dr. Robert Strengen Bigseth:

Your manuscript has been accepted, and I am forwarding it to the ASM Journals Department for publication. You will be notified when your proofs are ready to be viewed.

Sincerely,

Bonnie Prokesch
Editor, Microbiology Spectrum
